# US EPA EnviroAtlas Meter-Scale Urban Land Cover (MULC): 1-m Pixel Land Cover Class Definitions and Guidance

**Andrew Pilant [1],\*, Keith Endres [1], Daniel Rosenbaum [2] and Gillian Gundersen [3]**

1    MD243-05, Office of Research and Development, United States Environmental Protection Agency, Research Triangle Park, NC 27711, USA; endres.keith@epa.gov

2    Oak Ridge Institute for Science and Education, P.O. Box 117, Oak Ridge, TN 37831, USA; rosenbaum.daniel@epa.gov

3    Oak Ridge Associated Universities Inc., P.O. Box 117, Oak Ridge, TN 37831, USA; gillgundersen@gmail.com

\*    Correspondence: pilant.drew@epa.gov

**Abstract:** This article defines the land cover classes used in Meter-Scale Urban Land Cover (MULC), a unique, high resolution (one meter$^2$ per pixel) land cover dataset developed for 30 US communities for the United States Environmental Protection Agency (US EPA) EnviroAtlas. MULC data categorize the landscape into these land cover classes: impervious surface, tree, grass-herbaceous, shrub, soil-barren, water, wetland and agriculture. MULC data are used to calculate approximately 100 EnviroAtlas metrics that serve as indicators of nature's benefits (ecosystem goods and services). MULC, a dataset for which development is ongoing, is produced by multiple classification methods using aerial photo and LiDAR datasets. The mean overall fuzzy accuracy across the EnviroAtlas communities is 88% and mean Kappa coefficient is 0.84. MULC is available in EnviroAtlas via web browser, web map service (WMS) in the user's geographic information system (GIS), and as downloadable data at EPA Environmental Data Gateway. Fact sheets and metadata for each MULC community are available through EnviroAtlas. Some MULC applications include mapping green and grey infrastructure, connecting land cover with socioeconomic/demographic variables, street tree planting, urban heat island analysis, mosquito habitat risk mapping and bikeway planning. This article provides practical guidance for using MULC effectively and developing similar high resolution (HR) land cover data.

**Keywords:** high spatial resolution land cover data; remote sensing; EnviroAtlas; ecosystem services; decision support; image classification; machine learning; object-based image classification; rule-based image classification; pixel-based image classification; GIS; 1 m pixel

## 1. Introduction

Land cover (LC) data indicate the type, extent and configuration of the physical materials present at earth's surface (e.g., vegetation, built surfaces) and are essential to informed, effective stewardship of community landscapes, supporting decision making that integrates ecological, social, and economic factors. Toward this integration, the United States Environmental Protection Agency (US EPA) created EnviroAtlas (www.epa.gov/enviroatlas), a collection of interactive geospatial tools and resources that allows users to explore the many benefits people receive from nature, often referred to as ecosystem goods and services (EGS) [1]. Key components of EnviroAtlas are a multi-scaled interactive map, which provides easy access to EnviroAtlas data, the Eco-Health Relationship Browser, which shows linkages between ecosystems, the services they provide, and human health [2], and ecosystem services information and educational resources, including a range of lesson plans that educators may integrate into classrooms.

EnviroAtlas is organized at two spatial scales. A coarser national-scale component spans the conterminous US and builds on the US National Land Cover Dataset (NLCD) [3] with a 30 × 30 m pixel resolution. For a finer community-scale component, the EnviroAtlas team has developed Meter-Scale Urban Land Cover (MULC) at 1 × 1 m per pixel resolution, to support analysis and visualization of ecosystem services at a fine spatial resolution that captures individual trees, buildings and roads (Figures 1 and 2). For comparison, there are nine hundred MULC 1 × 1 m pixels to one NLCD 30 × 30 m pixel. A webmap of MULC examples can be found here: https://arcg.is/0fXjue0. As of 2020, there are 30 published EnviroAtlas MULC datasets: Austin, TX; Baltimore, MD; Birmingham, AL; Brownsville, TX; Chicago, IL; Cleveland, OH; Des Moines, IA; Durham, NC; Fresno, CA; Green Bay, WI; Los Angeles County, CA; Memphis, TN; Milwaukee, WI; Minneapolis/St. Paul, MN; New Bedford, MA; New Haven, CT; New York, NY; Patterson, NJ; Philadelphia, PA; Phoenix, AZ; Pittsburgh, PA; Portland, ME; Portland, OR; Salt Lake City, UT; Sonoma County, CA; St. Louis, MO; Tampa, FL; Virginia Beach, VA; Washington, D.C.; Woodbine, IA.

The term "meter-scale" indicates the general size range of the smallest identifiable features on the ground. This corresponds to objects approximately one to four meters in size. The size of the smallest detectable features varies, depending largely on the spectral and spatial contrast of the target against its background. Image quality, date and atmospheric conditions are also factors.

Similar high spatial resolution (HR) land cover (LC) data products have been developed by other groups, translated to MULC, and incorporated into EnviroAtlas. These external sources are the University of Vermont Spatial Analysis Lab, Sonoma Veg Map, the State of Iowa, Chesapeake Conservancy, Central Arizona-Phoenix (CAP LTER), Oneida Total Integrated Enterprises (OTIE), University of Arkansas Center for Advanced Spatial Technologies, and the Missouri Resource Assessment Partnership (MoRAP). After external LC data are translated to the MULC system, such data are considered equivalent to MULC, and are accompanied by the full suite of EnviroAtlas community EGS metrics. External land cover data sources, and how those data are translated to MULC, are specified in metadata for each MULC community.

As of 2010, approximately 81 percent of the United States (US) population lived in "urban areas" (US Census terminology for communities with population > 2500) [4]. Expanding urbanization is one motivation for developing high spatial resolution urban LC data for EnviroAtlas Communities. By modelling community landscapes at the fine MULC spatial scale of individual streets, buildings, trees, and lawns, we are better able to quantify landscape properties and patterns that contribute to human well-being and healthy urban ecosystems (Figures 1 and 2) and these EGS may then be better represented in making community decisions and policy. Potential MULC users include planning, commerce, transportation, recreation and public health authorities; water, wildlife and natural resource managers; community decision makers, teachers, students and citizens.

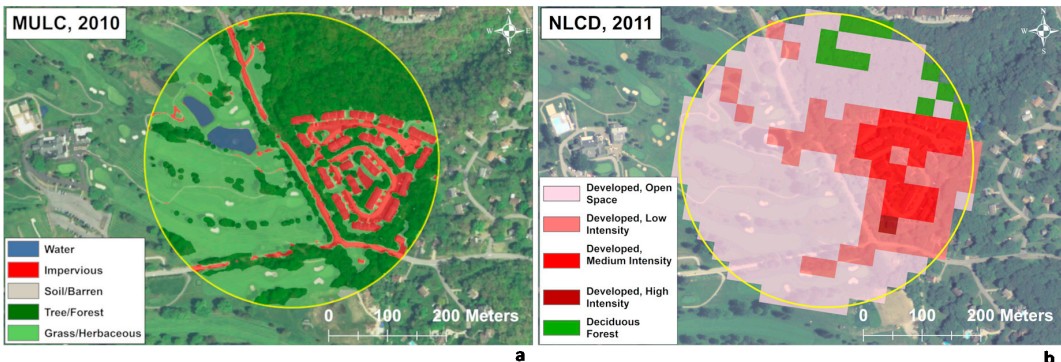

**Figure 1.** Comparison of spatial scale and level of detail of (**a**) Meter-Scale Urban Land Cover (MULC) (1 m pixel) and (**b**) National Land Cover Dataset (NLCD) (30 m pixel) in a Pittsburgh, PA neighborhood and golf course. Land cover 50% transparency over NAIP imagery.

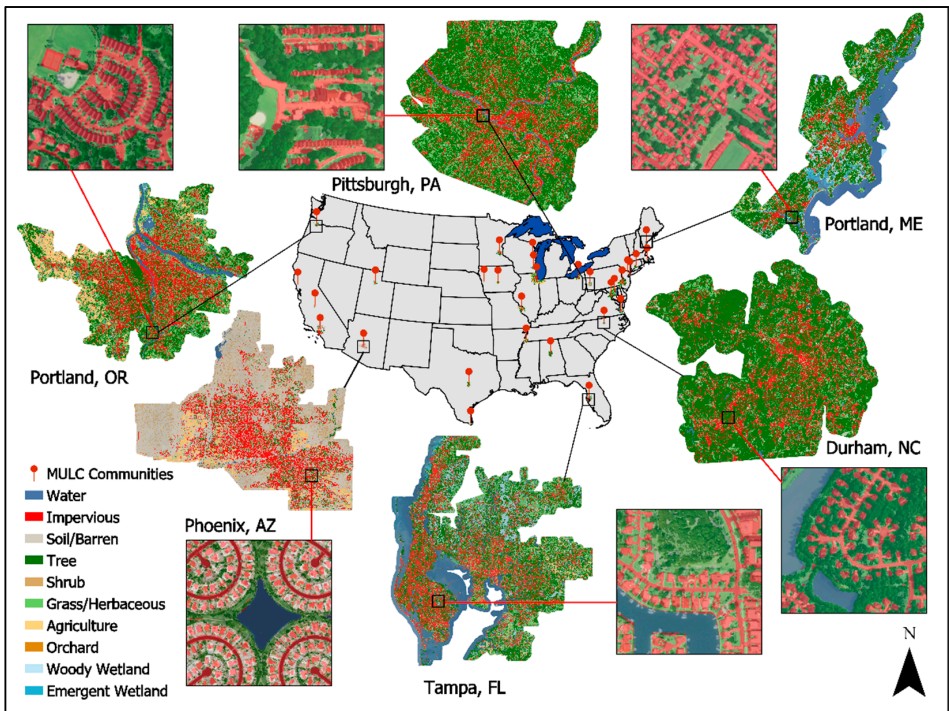

**Figure 2.** MULC examples for six of thirty EnviroAtlas communities. The six inset maps show MULC for each of the EnviroAtlas communities, with an expanded inset showing MULC at higher magnification. Community boundaries are from 2010 Census Urban Areas plus 1 km buffer.

The purpose of this paper is to define the EnviroAtlas MULC land cover classes, describe the processes used to generate MULC, and provide guidance to support the most effective use of MULC data. In the Materials and Methods section, we present the MULC design, aerial imagery and LiDAR data specifications, image classification methods and a fuzzy accuracy assessment method. Next, we define the MULC classes and their characteristics. The Results section summarizes statistics for 30 US EnviroAtlas communities. The Discussion section highlights some MULC applications and practical guidance for interpreting MULC data.

## 2. Materials and Methods

The MULC classes are intended to represent common urban landscape composition and features that can be reliably identified in 1 × 1 m pixels, visible near-infrared digital aerial photography, by human aerial photo interpreters, and by computer image classification algorithms. MULC classification design considerations include:

- encompass the LC types anticipated in US community landscapes;
- these LC classes are broadly recognized and understood by users;
- simplicity;
- the size of discrete landscape objects and features readily classifiable in single date, 1 × 1 m pixel imagery;
-  minimal confusion between classes;
- broad range of potential applications.

The Level 1 MULC classes are: water, impervious surface, soil-barren, tree, shrub, grass, agriculture and wetlands. To date, the more specific Level 2 classes have been used only as intermediate classes during the classification stage; they are provided for potential future analyses requiring more specificity. The shrub and agriculture classes are optional for a community, as discussed below.

MULC data span the 2010 to 2018 time period. Several datasets are circa 2010 to match the EnviroAtlas communities 2010 US Census focus. The date assigned to a community's MULC data is the predominant year of aerial imagery collection. Metadata indicate if multiple years of imagery are used, and year(s) of LiDAR data collection (which are typically not the same as the aerial imagery).

EnviroAtlas Community boundaries are defined by US Census Urban Area Block Groups for a Census Urban Area [5]. An additional 1 km buffer extends outward to eliminate potential edge effects when calculating EGS metrics based on moving-window analyses. In three cases, we have used county boundaries for data provided by external partners (Chicago, IL [comprised of 10 counties], Los Angeles, CA, and Sonoma County, CA). The county is a convenient geographic unit; it typically leverages coordinated geospatial, financial, and administrative resources.

### 2.1. Input Data

The input raster data stack for EPA-developed MULC typically consists of four-band aerial photography, normalized difference vegetation index (NDVI), and LiDAR data (height above ground and intensity layers) (Figure 3). Ancillary geospatial data layers (Table 1) are used as available, advantageous, and appropriate. to overlay agriculture and wetlands on the classified product, and for performing post-classification error correction.

Imagery from the United States Department of Agriculture (USDA) National Agriculture Imagery Program (NAIP) [6] is the primary MULC aerial photography. It has multiple advantages:

- high spatial resolution: 1 × 1 m pixels (and finer in some recent imagery);
- free (no cost) and available for most of the US;
- updates every two to three years by state;
- adequate horizontal positional accuracy (≤6 m by specification) (in our experience, NAIP is co-registered within about two meters of other HR image sources such as Google, Bing and ESRI);
- four spectral bands: three visible light bands (blue, green, red) and one near-infrared (NIR) band, which is used to derive a normalized difference vegetation index (NDVI).

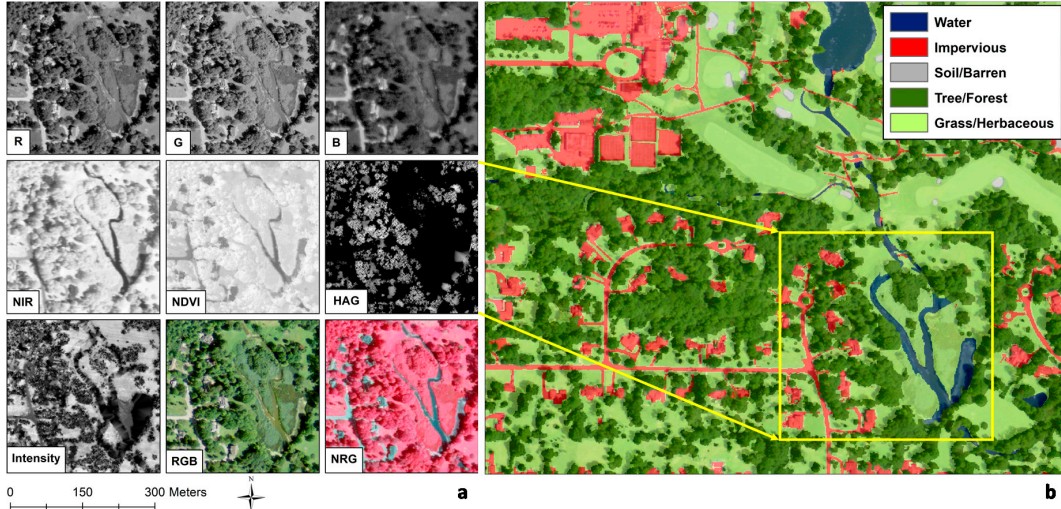

**Figure 3.** National Agriculture Imagery Program (NAIP) air photo and LiDAR raster band stack used in MULC classification. (**a**) Seven band raster stack typically used in MULC classification, plus RGB and NIR for reference. (**b**) MULC with 50% transparency overlaid on NAIP air photo showing region of zoomed insets in (**a**). R (red), G (green), B (blue), NIR (near infrared), NDVI (normalized difference vegetation index), HAG (LiDAR height above ground), Intensity (LiDAR intensity), RGB (red-green-blue true color), NRG (NIR-red-green false color composite). Note how different landscape features express differently between spectral bands.

NAIP imagery is acquired via internet download or external hard drives from the USDA Aerial Photography Field Office (APFO) or state sources. The standard data format is uncompressed 8 bit GeoTIFF with uncalibrated radiance represented by 256 grayscale levels in each band. Uncompressed data are used to retain maximum spatial and spectral fidelity needed in classification. NAIP images are typically tiled and distributed using the United States Geological Survey (USGS) 7.5 min quarter quadrangle topographic map series. A few MULC datasets adapted from external sources may use other HR aerial imagery as indicated in metadata.

While NAIP imagery is available across the entire conterminous US, airborne LiDAR data are not. We acquire LiDAR data as available from the USGS National Map [7], NOAA [8], and state and county geospatial data portals and personnel. LiDAR point clouds are interpolated into rasters of the following layers: digital elevation model (DEM) (all bare earth, ground points), digital surface model (DSM) (first point returns), height above ground (HAG) (also referred to as normalized DSM, or nDSM; HAG = DSM − DEM) and return pulse intensity. (Note: five initial EnviroAtlas communities—Durham, NC, New Bedford, MA, Paterson, NJ, Portland, ME, and Tampa Bay, FL—were produced without LiDAR).

### 2.2. Image Classification

EPA-developed MULC data are produced by classifying a raster dataset comprised of NAIP aerial photos, NDVI and LiDAR HAG and intensity data (Figure 3). Externally developed MULC datasets are classified from similar raster layers as specified in metadata. We have used three different classification approaches: pixel-based supervised, object-based supervised, and object rule-based. The approach used in each community is described in the metadata. Figure 4 shows the overall workflow for producing MULC data. Pixel-based and object-based classification methods are discussed in [9]. For MULC datasets created by supervised classification methods, training samples are selected from within the community boundary being mapped.

**Table 1.** Primary and supplemental data layers used to generate MULC.

| Acronym | Dataset Name | Comments/Usage |
| --- | --- | --- |
| NAIP (Primary data layer used in classification.) | National Agricultural Imagery Program | 1 m pixel, five band raster stack red-green-blue-near infrared-NDVI. Approximately three year update cycle. |
| LiDAR (Primary data layer used in classification.) | Light detection and ranging | 1 m pixel, two-band raster stack of LiDAR height-above-ground and intensity bands. |
| NLCD | National Land Cover Dataset | Inform algorithms and analysis, 30 m pixel size. Five year update cycle. |
| CCAP | NOAA Coastal Change Analysis Program | Inform algorithms and analysis, 30 m pixels. Recently, 1 to 5 m pixel data. Variable update cycle. |
| NHDPlus v2 | National Hydrography Data Plus Version 2. | Water and wetland feature vectors. Variable update cycle. |
| NWI | National Wetlands Inventory | Water and wetland feature vectors. Variable update cycle. |
| CLU | Common Land Units | Unattributed parcel polygons emphasizing agriculture. Vintage 2008. |
| CDL | Cropland Data Layer | Crop information at 30 m pixel size. USDA. Updated annually. |
| Roads and infrastructure | Road and utilities data layers | Vector data, as available. |
| Building footprint | Building footprint layers | Vector data, as available. |

The segmentation algorithms used in our object-based classification vary according to the software used: ArcGIS Desktop (v10.x) [10] and ArcGIS Pro (v2.x) (Segment Mean Shift) [11], ENVI (v5.x) (Watershed) [12], and eCognition (v9.x) (Multiresolution, Contrast Split) [13]. The analyst prepares to classify by studying existing land cover information to understand local vegetation, conditions and landscapes. NLCD and USDA Crop Land Data Layer [14] are particularly useful, in combination with HR imagery such as Google/Bing/ESRI satellite view, NAIP NIR, Google Street View and Bing Birdseye view.

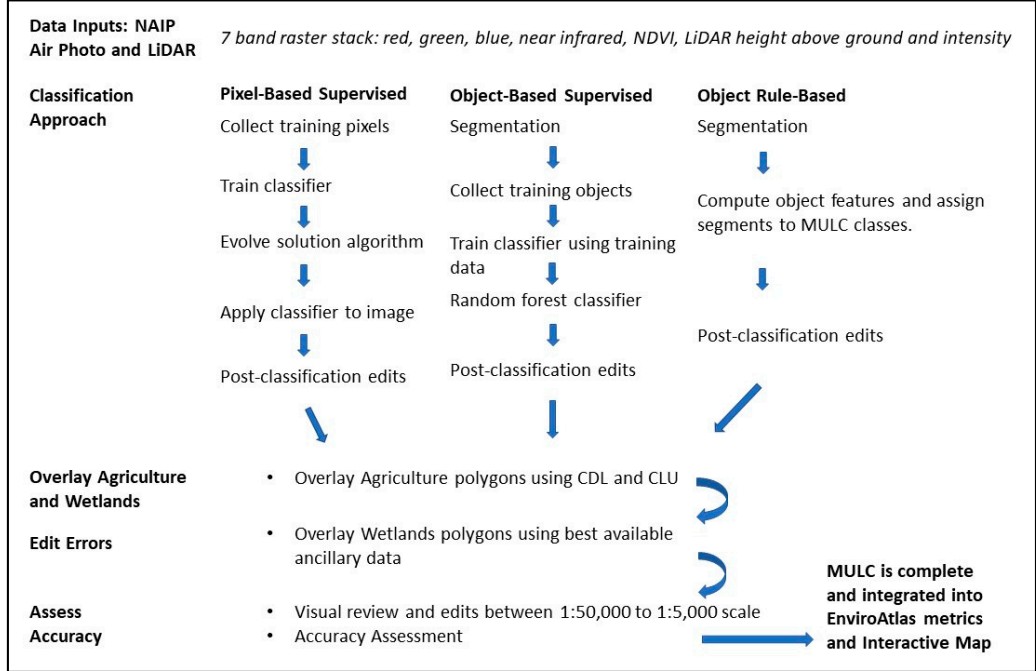

**Figure 4.** MULC classification workflow. MULC data are created using one of three classification methods. Abbreviations are defined in the text. Land cover datasets created by external partners are recoded to EnviroAtlas MULC classes and run through the post-classification steps (overlay, edit, assess).

*2.3. Post-Classification Operations*

2.3.1. Post-Classification Operations

The MULC data are reviewed after classification and errors are addressed in two ways. First, we perform as many edits as possible using GIS functions (e.g., conditional statements, convolution filtering). Ancillary spatial layers such as roads and building footprints are useful to mask and focus edits. We inspect each output layer to detect potential artifacts introduced by post-classification GIS functions. Second, we perform manual editing (on-screen, heads-up digitizing) to identify and recode remaining errors. Here the analyst interactively selects pixel groups (or polygons) for recoding from the incorrect to correct class. Manual editing is labor intensive and time consuming but can substantially improve the visual appearance.

2.3.2. Fuzzy Accuracy Assessment

We use a fuzzy approach [15] to assess the accuracy of the MULC classification. The motivation is to better accommodate the non-exclusive nature of land cover class membership: *"The need for using fuzzy sets arises from the observation that all map locations do not fit unambiguously in a single map category. Fuzzy sets allow for varying levels of set membership for multiple map categories. A linguistic measurement scale allows the kinds of comments commonly made during map evaluations to be used to quantify map accuracy"* [15].

An assessment analyst labels the land cover at each reference point (pixel) and assigns a fuzzy confidence value to the label on a scale from (1) (incorrect) to (5) (correct). For example, tree canopy over grass is a situation where both classes could be considered "correct". The sensor may capture both the canopy and the ground through thin canopy or canopy gaps. The analyst might assign a tree label with confidence of 4, and a grass label with a confidence of 3. Another situation is accommodating the continuum between grass and soil endmembers. The fuzzy approach allows both agriculture and soil class labels to be considered correct for a barren crop field. Accuracy assessment results are presented in two confusion (error) matrices, showing errors of omission (producer's accuracy) and

errors of commission (user's accuracy) for each class as well as an overall accuracy value. The fuzzy confusion matrix is less conservative and based on these fuzzy confidence evaluations; the non-fuzzy confusion matrix is more conservative and based on strict binary correct/incorrect class membership.

The MULC classification is compared to 500 to 700 randomly distributed photo interpreted reference points (i.e., an initial target of 100 reference points per class, 5 to 7 classes per community). If rare classes (e.g., soil, water) are under-sampled (n < 50), additional reference points are collected to reach n ≥ 50, stratified by class as indicated by the MULC classification. The NAIP imagery input to the classification serves as the primary photo interpreted reference imagery. This assures spatial and temporal correspondence of the reference imagery and the MULC classification. Uninterpretable or ambiguous points may be removed from consideration (e.g., deep shadow or boundary between classes). Photo interpretation is aided by spatially linked displays of LiDAR-derived layers, NIR false color composite and other temporally appropriate high resolution imagery as noted above. Wetland classes (woody and emergent) are not included in the accuracy assessment. Because remote identification of wetlands is complex and beyond the scope of our study, we assume that the ancillary wetlands data are reliable. However, reference pixels located in wetland areas are assessed in terms of their non-wetlands, underlying MULC class.

The final quality assurance step is on-screen visual assessment of the classified MULC by multiple analysts at scales from 1:50,000 to 1:5000. Known errors and uncertainties are described in the metadata for each community.

*2.4. Definitions of MULC Classes*

The standard EnviroAtlas MULC product is provided at a "Level 1" thematic resolution and is similar but not identical to the Anderson and NLCD Level 1 classes (Table 2) [3,16]. MULC data are published at Level 1; a structure of Level 2 classes is provided below in anticipation of potential future analyses requiring greater thematic specificity.

As discussed above, data are either created by EPA EnviroAtlas personnel or incorporated from external non-EPA sources. Externally produced data must meet these criteria:

- The classes can be unambiguously translated to the MULC system;
- The data are at same or finer spatial resolution;
- The data are sufficiently contemporaneous with the EnviroAtlas period of study;
- The data have an overall target fuzzy accuracy ≥80%. (We perform the standard MULC accuracy assessment on externally developed LC data.)

To the first point, a dataset acquired from external sources that contains separate building and street classes can be unambiguously recoded into the MULC Impervious Surface class. However, a hypothetical residential class defined as "50% impervious surface and 50% vegetation" cannot be used because impervious, trees, shrubs and grass are inseparably combined into a single class and cannot be unmixed.

**Table 2.** MULC Level 1 and 2 class names, codes, and descriptions.

| Standard MULC Level 1 Class | Level 1 Code | Level 2 Codes | Short Description |
|---|---|---|---|
| Water | 10 Water | 11 Fresh Water<br>12 Salt Water<br>13 (available)<br>14 Drinking Water Reservoir | The water class includes all natural and some anthropogenic surface waters: rivers, streams, canals, ponds, reservoirs, lakes, bays, estuaries, and near-shore coastal waters. |
| Impervious Surface | 20 Impervious Surface | 21 Dark Impervious (low reflectance)<br>22 Light Impervious (high reflectance)<br>23 Road<br>24 Building<br>25 Parking Area<br>26 Soil and Gravel Impervious<br>27 Solar Panel | The impervious class includes buildings, paved roads, parking lots, driveways, sidewalks, roofs, swimming pools, patios, painted surfaces, wooden structures, solar farms and most asphalt and concrete surfaces. Swimming pools, and wastewater treatment tanks and basins, are labeled as Impervious as described in the text. |
| Soil-Barren | 30 Soil and Barren Land | 31 Developed Soil (soil in developed areas likely compacted and disturbed)<br>32 Bare Rock<br>33 Sand and Gravel<br>34 Quarry | The soil and barren class ("soil") includes bare soil, bare rock, mud, clay, sand, barren agricultural fields (for communities with less than 5% agriculture), construction sites, quarries, gravel pits, mine lands, industrial land, parking lots, golf course sand traps, ball parks, playgrounds, stream and river sand bars, sand dunes, beaches and other bare soil and gravel surfaces. |
| Tree | 40 Tree | 41 Deciduous Tree/Forest<br>42 Evergreen Tree/Forest<br>43 Mixed Tree/Forest<br>45 Low Tree (height < 2 m)<br>46 Medium Tree (2 m ≤ height < 5 m)<br>47 High Tree (height ≥ 5 m) | Woody single stem vegetation ≥ 2 m height. "tall vegetation." Generally, the branching starts above a specified trunk height, in contrast with shrubs where branching starts near ground level. Classes 45–47 are optional and subordinate to height thresholds defined in the text. The term Tree comprises trees of varying extent: individual trees, stands and forest. The codes 41–47 are provided as guidelines for potential future analyses; they have not been used in MULC data to date. |
| Shrub | 50 Shrubs or Shrubland | 51 Shrubland or Scrubland (undifferentiated Shrub, Soil and Grass)<br>52 Individual Shrub in Natural Environment<br>53 Individual Shrub in Built/Developed Environment.<br>55 Low Shrub (height ≤ 2 m)<br>56 Medium Shrub (2 m < height < 5 m)<br>57 High Shrub (height ≥ 5 m) | Woody multiple stem vegetation with height ≤ 2 m and > 0.5 m. "medium height vegetation." Shrub mapping requires LiDAR height above ground except in known shrub land areas. |
| Grass-Herbaceous | 70 Grass | 71 Lawns and Other Managed Grass<br>72 Roadside Grass<br>73 Pasture<br>74 Natural Grassland (e.g., prairie) | Graminoids, forbs and herbs lacking persistent woody stems; includes residential lawns, golf courses, roadway medians and verges, park lands, transmission line and natural gas corridors, recent forest clear-cuts, meadows, pasture, grasslands and prairie grass. Also known as "low vegetation." Grass classified in wetlands areas is recoded to emergent wetlands. |

**Table 2.** *Cont.*

| Standard MULC Level 1 Class | Level 1 Code | Level 2 Codes | Short Description |
|---|---|---|---|
| Agriculture | 81 Agriculture | 80 Row Crops<br>82 Orchard | Row crops (80) and orchards (82) (Note: agriculture class numbering departs from the norm.) Pixels classified as grass (70) are recoded to row crop (80) when the ancillary agricultural polygons are overlaid. |
| Wetlands | 90 Wetlands | 91 Emergent Wetland<br>92 Woody Wetland | Emergent (91) and woody (92) wetlands Wetlands polygons are overlaid on classified MULC using best available data (e.g., NWI, NHD+). Grass recodes to emergent wetland. Trees recode to woody wetland. Soil, water and impervious classes remain unchanged. Treatment of shrubs is indicated in metadata. |

### 2.4.1. Unclassified

The 00 unclassified class is available for special cases or unanticipated LC classes not present in the existing MULC system.

### 2.4.2. Water

The water class includes all natural and some anthropogenic surface waters: rivers, streams, canals, ponds, natural lakes, artificial lakes, dammed valley reservoirs, bays, estuaries and near-shore coastal waters. Note that wastewater treatment tanks, clarifiers, basins and sumps are labeled impervious surfaces, as are swimming pools, fountains and similar small anthropogenic water features. This distinction is made based on their ecosystem services which are very different to those in the forms of surface water above: they are not biologically active (wastewater treatment notwithstanding); they are closed systems without natural surface water exchange with the environment; they are constructed features. The water class is most commonly confused with shadow, trees and dark impervious surfaces. Bright sun glint on water is confused with highly reflective classes such as soil or impervious surface. Turbid, sediment-laden brown or tan water is confused with soil. Shallow water is confused with soil, impervious and vegetation depending on bottom surface optics of the substrate (e.g., sand, silt, rock, submerged vegetation). Water with floating vegetation may misclassify as vegetation but is intended to be in the water class. Floating vegetation is assumed to be ephemeral, and that the LC at such a point is better represented as water than vegetation.

Lakes, ponds, tidal zones, estuaries and other water bodies that vary in extent and shoreline location over time are mapped according to how they appear in the imagery; i.e., at the date and time of image acquisition. If circumstances favor using a different shoreline (e.g., authoritative NOAA shoreline) this is indicated in the metadata.

### 2.4.3. Impervious Surface

An impervious surface prevents or substantially limits rainfall and other water from infiltrating into the soil. The impervious class includes paved roads, parking lots, driveways, sidewalks, roofs, swimming pools, patios, painted surfaces, wooden structures and most asphalt, concrete and paved surfaces. In MULC, dirt roads, gravel roads and railways are classified as impervious. These areas are compacted, disturbed and altered leading to a loss of perviousness. Except for bare rock, most impervious surfaces are anthropogenic and most pervious surfaces are natural (e.g., vegetation, soil). Bare rock is functionally impervious and is commonly confused spectrally with the impervious class, but in MULC it is assigned to the soil and barren class. Rooftops and roads that incorporate sand and clay materials are spectrally confused with soil but belong in the impervious class.

Level 1 MULC combines roads/pavements and buildings into one impervious class (20), rather than separate Level 2 roads (23) and buildings (24) because of the requirement for height information to classify buildings. The original MULC classes are designed to be classified from NAIP imagery, with or without LiDAR, because of patchy LiDAR availability. If height above ground or building footprints are available, one can separate roads and buildings.

Solar panel farms (class 27) are a separate impervious Level 2 class. They represent a third type of impervious built surface after pavements and buildings/rooftops. Solar panels present an interesting EGS case in that biological functions continue beneath the artificial canopy. The panels provide shade and collect and distribute rain preferentially.

### 2.4.4. Soil-Barren

The soil and barren class ("soil") includes soil, bare rock, mud, clay, sand, barren (fallow) agricultural fields, construction sites, quarries, gravel pits, mine lands, recreational areas, golf course sand traps, ball parks, playgrounds, stream and river sand bars, sand dunes, beaches and other bare soil, sand, gravel and rock surfaces. Soil and barren includes natural areas with widely spaced or no vegetation cover, including the soil substrate of semiarid and arid rangeland, shrubland and desert. Unpaved dirt roads, gravel roads, and railways are typically semi-impervious, and are assigned to the impervious class unless otherwise noted.

Soil is a relatively rare class in humid temperate communities such as Milwaukee, WI, Pittsburgh, PA and Portland, ME. Soil is more common in arid communities such as Phoenix, AZ and Fresno, CA. Construction sites are a common soil surface in highly developed urban landscapes, and barren agricultural fields on the periphery. Soil is commonly confused with light impervious surfaces.

### 2.4.5. Tree

The tree class includes trees of any kind, from a single individual to continuous canopy forest. Trees are single stem woody perennial plants with a trunk, branches and leaves and height greater than 2 m. Signature characteristics of the tree class in NAIP imagery include greenness, high NIR reflectance, NDVI, a mottled textured canopy, tree crowns illuminated and shadowed on opposite facets, visible trunks, length of shadows and context. Signature characteristics of the tree class in LiDAR include height above ground, intensity, object shape, multiple LiDAR returns and canopy surface texture.

Level 1 MULC combines deciduous and evergreen trees in one tree class. Shrubs greater than two meters height are classified as tree unless otherwise indicated. Bamboo is botanically a grass (family Poaceae) but is classified as tree here if height ≥ 2 m. Trees are most commonly confused with water, dark impervious, shrub and grass.

Tree canopy pixels that extend over other LC surfaces such as streets, buildings and lawns are assigned to tree rather than the underlying class. The tree canopy is what the sensor "sees" in its direct line of sight. This convention reflects an EnviroAtlas emphasis on EGS and the importance of street trees in urban areas. Thus, where trees extend over a road, driveway, sidewalk or rooftop, the amount of underlying impervious surface (or grass, soil or water) will be underestimated. The horizontal surface area of tree canopy will be correctly estimated. If accurate road and building footprint data are available, one may compute the under-canopy extent of these obscured surfaces.

### 2.4.6. Shrub

Shrubs are multiple stem woody perennial plants between 0.5–2 m height. Shrubs are recognized in air photos by context (e.g., desert, rangeland, urban landscaping), the mottled texture of the canopy (compared to grass), and lesser shadows (compared to trees). Shrubs are recognized by height (and possibly shape) in LiDAR data.

In some land cover datasets, arid and semiarid natural shrub vegetation is mapped as undifferentiated shrubland (51). In that case, shrubs, soil, and grass are mixed in a single class, rather than as differentiated classes of shrub (52), grass (70) and soil (30). In EnviroAtlas, using shrub

class (52) (individual shrubs) is preferred over shrubland (51). Shrub (52) is at a finer information granularity to support EGS analysis.

### 2.4.7. Grass-Herbaceous

The grass and herbaceous class ("grass") includes the graminoids, forbs and herbs lacking persistent woody stems. Grass includes residential lawns, golf courses, roadway medians and verges, park lands, transmission line and natural gas corridors, recently clear-cut forest areas, pasture, grasslands, and prairie grass. Small shrubs may fall into this category as noted above. It is also known as "low vegetation."

For healthy, photosynthetically active grass, the principal identifying characteristics in NAIP imagery are greenness, high reflectance in the near infrared, high NDVI, urban context and a smoother image texture than tree and shrub canopy. Context helps in identifying grass (e.g., proximity to a building, athletic field or highway). NAIP imagery is collected in summer leaf-on conditions when grass may be green, or brown with heat and moisture stress. Sparse or brown grass is commonly confused with soil and impervious surfaces. Grass-soil confusion is greater in arid than in humid-temperate environments.

What to do with indeterminate pixels in NAIP imagery that could be either grass or soil? Sparse, brown or dead grass are spectrally like soil, and soil and grass intermix along a continuum. As a guideline, if potential grass or soil pixels/polygons show above-background reflectance in the near-infrared band (indicative of photosynthetic activity), they are labeled grass. An operational assumption is that, except in arid regions, soil has the potential to support grass or other vegetation at some point during the growing season. The analyst consults other HR imagery from different dates to assess if grass is present at other times.

### 2.4.8. Agriculture

Agriculture is a layer superimposed on the MULC classification. The USDA Common Land Unit (polygon) [17] and raster Cropland Data Layer (CDL) [14] are used to help identify agriculture polygons. Level 2 Agriculture is labeled as row crops (80) if MULC pixels are classified as grass, shrub, or soil and fall within these ancillary agricultural datasets, and orchards (82) if classified as tree. (Note: the agriculture class numbering deviates slightly from standard MULC class numbering conventions due to a transcription error in the initial data upload). Pasture is assigned to the grass class for two reasons: (1) difference in land management practices between row crops and pasture, and (2) difficulty differentiating pasture from non-cultivated grass.

The agriculture ("Ag") class is included in a MULC product if the most recent NLCD indicates agriculture greater than 5% within the EnviroAtlas community boundary. If agriculture is less than or equal to 5%, agriculture pixels (polygons) are labeled as whatever LC is on the ground when the NAIP imagery is acquired (grass, soil, shrub or tree), rather than as agriculture. Twenty of the EnviroAtlas communities have an agriculture class.

### 2.4.9. Wetlands

As defined by Section 404 of the Clean Water Act: "Wetlands are areas that are inundated or saturated by surface or ground water at a frequency and duration sufficient to support, and that under normal circumstances do support, a prevalence of vegetation typically adapted for life in saturated soil conditions" [18]. Wetlands include swamps, marshes, bogs, and other wet and flooded areas [19,20]. Like agriculture, in MULC data, wetlands are delineated using the best available ancillary data, which to date have been the U.S. Fish and Wildlife Service National Wetlands Inventory (NWI) [21] and U.S.G.S. National Hydrography Dataset (NHDPlus v2) [22]. Classifying wetlands directly from imagery/LiDAR is beyond the scope of this study, and generally requires ground validation and ancillary data. Wetlands boundary polygons are overlaid on the MULC data; areas classified as tree are labeled woody wetland (91), and areas classified as grass-herbaceous are labeled emergent wetland

(92). Treatment of shrub areas is indicated in the community metadata. Visual checks are performed for thematic and positional agreement of wetlands layers and underlying imagery.

## 3. Results

Here we present statistics characterizing the MULC dataset. Table 3 summarizes the size, population, year and accuracy statistics for 30 EnviroAtlas communities. MULC communities range considerably in both aerial extent and population, with the largest community (Chicago) encompassing more than 14,000 km$^2$ and the smallest (Paterson, NJ) spanning just 47 km$^2$. The mean area is 3139 km$^2$. Community populations closely aligned with aerial extent in a positive relationship. The largest community population is over 9.8 million (Los Angeles, CA County, 11,336 km$^2$) and the smallest is just over 1500 people (Woodbine, IA, 51 km$^2$). The mean population for EnviroAtlas communities is 2.1 million people according to the 2010 U.S. Census [23].

**Table 3.** MULC statistics for EnviroAtlas communities. Abbreviations after the community name indicate the main method used in classification: pixel-based supervised (PBS), object-based supervised (OBS), or object rule-based (ORB).

| EnviroAtlas Community | Area (km$^2$) | Population (2010 Census) | Overall Accuracy (Fuzzy) | Kappa (Fuzzy) | Overall Accuracy (Non-Fuzzy) | Kappa (Non-Fuzzy) | Imagery Date | LiDAR Dates |
|---|---|---|---|---|---|---|---|---|
| Austin, TX (PBS) | 2499 | 1,334,516 | 90.7 | 87.9 | 86.5 | 82.6 | 2010 | 2007 |
| Baltimore, MD (ORB) | 4545 | 2,252,753 | 92.7 | 90.5 | 90.1 | 87.1 | 2013 | 2004, 2005, 2011, 2015 |
| Birmingham, AL (PBS) | 2335 | 763,628 | 87.6 | 80.5 | 83.4 | 74.2 | 2011 | 2010, 2011, 2013 |
| Brownsville, TX (PBS) | 938 | 223,572 | 82.3 | 77.6 | 76.7 | 70.6 | 2014 | 2011, 2006 |
| Chicago, IL (ORB) | 14,687 | 9,203,201 | 86.8 | 83.6 | 80.8 | 76.0 | 2010, 2012, 2013 | 2006, 2007, 2008, 2010, 2013, 2014 |
| Cleveland, OH (PBS/ORB) | 2737 | 1,758,114 | 90.2 | 86.9 | 86.2 | 81.6 | 2011, 2013 | 2006 |
| Des Moines, IA (PBS) | 1130 | 456,017 | 84.6 | 80.4 | 77.6 | 73.5 | 1 m for 2008, 2009; 0.61 m for 2007, 2009, 2010 | 2009 |
| Durham, NC (PBS) | 569 | 340,851 | N/A | N/A | 83.0 | 78.8 | 2010 | N/A |
| Fresno, CA (PBS) | 753 | 659,628 | 86.9 | 83.5 | 81.1 | 76.2 | 2010 | 2012 |
| Green Bay, WI (PBS) | 857 | 219,947 | 94.1 | 92.7 | 90.4 | 87.9 | 2010 | 2010 |
| Los Angeles County (ORB) | 11,336 | 9,818,599 | 89.2 | 86.2 | 61.1 | 53.4 | 2014, 2016 | 2016 |
| Memphis, TN (PBS) | 2516 | 1,091,638 | 89.0 | 86.1 | 86.9 | 83.5 | 2012, 2013 | 2009, 2010, 2011, 2012 |
| Milwaukee, WI (ORB) | 2154 | 1,373,711 | 85.5 | 80.5 | 76.2 | 68.9 | 2010 | 2010 |
| Minneapolis/St. Paul, MN (ORB) | 3085 | 2,282,061 | 87.7 | 84.4 | 87.1 | 83.6 | 2010 | 2011 |
| New Bedford, MA (PBS) | 258 | 151,164 | 95.0 | 93.0 | 92.3 | 89.2 | 2010 | N/A |
| New Haven, CT (PBS) | 1422 | 578,536 | 92.0 | 88.8 | 89.0 | 83.2 | 2014 | 2006, 2010, 2011 |

**Table 3.** *Cont.*

| EnviroAtlas Community | Area (km²) | Population (2010 Census) | Overall Accuracy (Fuzzy) | Kappa (Fuzzy) | Overall Accuracy (Non-Fuzzy) | Kappa (Non-Fuzzy) | Imagery Date | LiDAR Dates |
|---|---|---|---|---|---|---|---|---|
| New York, NY (ORB) | 1109 | 8,175,131 | 87.4 | 83.2 | 84.2 | 79.0 | 2011 | 2010 |
| Paterson, NJ (PBS) | 47 | 146,199 | 92.5 | 89.2 | 86.8 | 81.2 | 2010 | N/A |
| Philadelphia, PA (ORB) | 7184 | 5,425,378 | 86.4 | 82.3 | 77.6 | 70.6 | 2005–2008, 2012–2015 | 2006–2008, 2011–2015 |
| Phoenix, AZ (ORB) | 5406 | 3,704,874 | 75.4 | 65.9 | 69.2 | 57.7 | 2010 | N/A |
| Pittsburgh, PA (PBS) | 1927 | 1,209,128 | 89.3 | 85.1 | 86.5 | 81.3 | 2010 | 2006 |
| Portland, ME (PBS) | 523 | 191,292 | N/A | N/A | 87.5 | 85.0 | 2010 | N/A |
| Portland, OR (PBS) | 2507 | 1,853,233 | 91.4 | 89.2 | 78.5 | 73.5 | 2011, 2012 | 2007, 2010 |
| Salt Lake City, UT (ORB) | 2244 | 1,030,599 | 82.5 | 77.0 | 78.7 | 72.0 | 2014 | 2006–2007, 2011, 2013–2014 |
| Sonoma County, CA (OBS) | 4910 | 483,878 | 80.9 | 75.8 | 79.0 | 73.4 | 2011,2013 | 2013 |
| St. Louis, MO (OBS) | 4188 | 2,174,437 | 90.4 | 87.9 | 82.3 | 77.7 | 2012, 2014–2016 | 2008–2010, 2012 |
| Tampa, FL (PBS) | 4492 | 2,517,798 | N/A | N/A | 70.6 | 65.2 | 2010 | N/A |
| Virginia Beach, VA (OBS) | 3255 | 1,541,779 | 84.1 | 80.6 | 83.5 | 79.9 | 2013, 2014 | 2015, 2010, 2013 |
| Washington, DC (ORB) | 5423 | 4,693,748 | 91.5 | 88.7 | 85.4 | 80.6 | 2013, 2014 | 2004, 2008, 2011, 2012, 2015, 2016 |
| Woodbine, IA (PBS) | 51 | 1555 | 90.2 | 84.4 | 87.0 | 79.3 | 2011 | 2009 |
| Mean | 3170 | 2,188,566 | 88.0 | 84.0 | 82.0 | 77.0 | | |
| Total | 95,088 | 65,656,965 | | | | | | |

N/A means data not available for this community. Fuzzy accuracy assessment was implemented after the first three communities were created and not retroactively performed.

The data used for classification in each community vary by availability, and typically the most recent available data are prioritized. Twelve of the 30 communities published to EnviroAtlas are based on 2010 NAIP imagery and most of the other communities are based on NAIP imagery from 2016 or earlier (Table 3). Twenty-four community datasets incorporate LiDAR, but, due to the timing of LiDAR acquisition, only four of those communities have LiDAR matching the imagery collection dates. LiDAR is not collected as frequently as imagery, and collection years for both LiDAR and imagery often do not overlap. When imagery and LiDAR are of different dates and do not agree due to land use changes, the analyst typically defers to the imagery when making post-classification corrections and during the accuracy assessment process. Data limitations for each community are indicated in metadata.

Table 4 summarizes fuzzy and non-fuzzy MULC accuracies by class for all existing EnviroAtlas communities. Table 5 is a confusion matrix constructed from 17,760 reference points for the 27 EnviroAtlas communities that have received both fuzzy and non-fuzzy accuracy assessments, illuminating the nature of interclass confusion.

**Table 4.** MULC mean class accuracies across 30 EnviroAtlas communities.

|  | Agriculture | Grass | Impervious | Shrub | Soil | Tree | Water |
|---|---|---|---|---|---|---|---|
| Fuzzy User Accuracy | 90.4 | 82.6 | 90.8 | 79.7 | 76 | 86.9 | 96.1 |
| Fuzzy Producer Accuracy | 94.8 | 78.5 | 85.7 | 80.9 | 82.4 | 88.6 | 95.7 |
| Non-Fuzzy User Accuracy | 82.5 | 75.1 | 87.8 | 71.6 | 63.2 | 83.5 | 95.3 |
| Non-Fuzzy Producer Accuracy | 82.6 | 68.7 | 83.3 | 71.7 | 73.8 | 85.3 | 94.2 |

Class user and producer accuracies for all communities are generally high and increase between non-fuzzy and fuzzy assessments (Table 4). The class user accuracy, calculated by dividing the number of correct reference points (where both the row and column classes agree) for a class by the row total, indicates how well the land cover represents the class as defined by the reference points. The class producer accuracy, calculated by dividing the number of correct reference points for a class by the column total, indicates how well the class is represented in the classification. The most accurate class in MULC landcover is water. Twenty-eight communities have both fuzzy and non-fuzzy accuracy assessments. The mean overall accuracy across all EnviroAtlas communities is 88% fuzzy and 82% non-fuzzy. Overall fuzzy accuracy is always higher than overall non-fuzzy accuracy. Mean kappa values are 0.84 fuzzy and 0.77 non-fuzzy. The soil class has the lowest user accuracy (77.8%) and grass class has the lowest producer accuracy (78.9%). Based on the fuzzy confusion matrix (Table 5), grass class confusion is mostly with soil and tree classes.

**Table 5.** Fuzzy confusion matrix for n = 27 communities.

| | | Reference Classes | | | | | | | | |
|---|---|---|---|---|---|---|---|---|---|---|
| | | Agriculture | Grass | Impervious | Shrub | Soil | Tree | Water | Row Total | User Accuracy |
| | Agriculture | 1771 | 108 | 4 | 1 | 39 | 10 | 1 | 1934 | 0.92 |
| | Grass | 19 | 3100 | 192 | 16 | 98 | 280 | 15 | 3720 | 0.83 |
| Land cover Classes | Impervious | 2 | 111 | 2761 | 2 | 86 | 59 | 8 | 3029 | 0.91 |
| | Shrub | 3 | 46 | 20 | 441 | 24 | 11 | 0 | 545 | 0.81 |
| | Soil | 27 | 198 | 117 | 23 | 1446 | 24 | 24 | 1859 | 0.78 |
| | Tree | 5 | 351 | 129 | 29 | 21 | 4359 | 13 | 4907 | 0.89 |
| | Water | 0 | 17 | 8 | 0 | 22 | 12 | 1707 | 1766 | 0.97 |
| | Column Total | 1827 | 3931 | 3231 | 512 | 1736 | 4755 | 1768 | 17760 | |
| | Producer Accuracy | 0.97 | 0.79 | 0.86 | 0.86 | 0.83 | 0.92 | 0.97 | | |
| | Overall Fuzzy Accuracy | 0.88 | | | | | | | | |
| | Kappa | 0.85 | | | | | | | | |

Three early communities are omitted because they lack a fuzzy accuracy assessment: Durham, NC; Portland, ME; Tampa, FL.

Soil class mixing is mostly with grass and impervious. Shrub landcover class is mapped in only six (western) communities and consequently has fewer reference points than the other classes.

## 4. Discussion

MULC underpins metrics that complement those derived from the national, 30 m resolution land cover component in EnviroAtlas, offering decision makers, researchers and others the ability to evaluate ecosystem services and land cover characteristics at household/street, community, neighborhood (block group), city, and regional levels. MULC and other EnviroAtlas data have been used in a range of

applications from regional to local scales. In Portland, Oregon, city planners have used MULC to design street tree planting and green infrastructure for urban heat island mitigation [24]. In Durham, North Carolina, EnviroAtlas has been used to identify census block groups with low tree cover and vulnerable populations to explore how tree planting might benefit child development, overall public health, and environmental quality [25,26]. In Tampa Bay, Florida, a Health Impact Assessment (HIA) has demonstrated how MULC and EnviroAtlas metrics, tools, and data can assist decision makers in a health and wellness application [27]. See the EnviroAtlas use case page for more information (https://www.epa.gov/enviroatlas/enviroatlas-use-cases).

The high level of detail provided by MULC data contributes to diverse research, ranging from environmental and public health to the economic benefits attributed to EGS. MULC data and derived EnviroAtlas community metrics support research including epidemiological studies on the salutogenic effects of natural environment exposure in urban areas [28], mosquito distribution analyses to assess vector-borne disease risk in Texas [29], and urban revitalization efforts in the Great Lakes region [30], among others. A bibliography of research using MULC and other EnviroAtlas data can be found on the EPA EnviroAtlas website: https://www.epa.gov/enviroatlas/enviroatlas-publications.

### 4.1. Uncertainty in MULC Data

In this section, we discuss MULC interpretation and origins of common uncertainties in MULC data. It is important that map classification errors be understood so that the EGS metrics can be accurately estimated, as major map errors can translate into incorrect valuation of ecosystems [31]. A confusion matrix is just one expression of map accuracy and users have varying needs that may prioritize map characteristics other than the statistical evaluation of reference points. MULC datasets have been developed using both pixel-based and object-based methods and land cover features, in actuality, are groupings of pixels representing the real world. MULC users who reside in the communities represented in the MULC dataset series are likely to possess the best understanding of accurate (or expected) land cover types in the areas of interest. There are times when a map can have high statistical accuracy but still possess errors in a particular area of local interest; this can make users lose confidence in the product. It is important that a map has good statistical accuracy, but also accurately represents real conditions for local users. It is for that reason that we spend a large portion of the data development process in quality assurance (QA) to ensure that MULC datasets possess acceptable statistical accuracies and have minimal visual errors.

### 4.2. Evaluation and Uncertainty in MULC

We recommend that to evaluate MULC data, the user display MULC at 40–60% transparency overlaid on the source imagery (e.g., NAIP) basemap and view at multiple zoom levels. This allows direct comparison of the MULC layer and source imagery. Comparing with higher resolution (e.g., 0.1–0.5 m) imagery may add additional useful information. Displaying the MULC over a more recent image basemap may visually highlight sites of land cover change.

There are multiple types of uncertainty and errors in high resolution land cover data for consideration when evaluating data accuracy and quality:

1. True misclassification (e.g., the image pixel is composed of soil but the map labelled it grass);
2. Non-exclusive class membership (the pixel is a mixture of soil and grass);
3. Inter-observer error (the map developer and the accuracy assessor use different criteria (e.g., pixel color and brightness) for labeling an ambiguous pixel as either soil or grass)
4. Severity of misclassification (for a specific user application, mistaking grass for tree may be less significant than mistaking soil for tree [i.e., vegetated versus non-vegetated] or water for impervious);
5. Items 2 and 3 are allowed greater flexibility as a result of the fuzzy accuracy assessments employed for MULC datasets.

Sources of errors and uncertainty in MULC data include: ambiguous class membership (e.g., the grass-soil continuum); shadows; image quality and dynamic range; radiometry and solar geometry differences between image acquisition dates for a community; different image and LiDAR acquisition dates; low quality or missing LiDAR data (e.g., non-returns over water); errors or misalignment of ancillary data.

### 4.3. Grass-Soil Confusion

Grass and soil show the greatest class confusion (Tables 4 and 5). A pixel may be on a continuum between all grass and all soil, and brown, senescent grass is spectrally and texturally similar to soil. Another factor is potential differences in heuristic thresholds used by the MULC data developer and accuracy assessor to distinguish grass from soil. There are ambiguous or borderline cases for distinguishing grass and soil, especially in semi-arid and arid locales; e.g., Los Angeles, CA and Phoenix, AZ. When grass is brown, sparse, stressed or senescent, the green and near infrared reflectance are reduced and more resemble a soil spectral signature. NAIP data are collected in summer when non-irrigated vegetation may be brown and water stressed. Because grass and soil land cover have different ecosystem services and functions, we strive to differentiate them in a community's MULC. An EnviroAtlas MULC convention is to assume that most soils (especially in more humid ecoregions) are capable of supporting some amount of grass or other low vegetation, so the classification algorithms are tuned to favor a grass label under ambiguous circumstances. The MULC developer examines ancillary aerial and street imagery from multiple dates to see if grass is present at other times of the year to determine algorithm thresholds for discriminating grass from soil.

### 4.4. Soil-Impervious Confusion

Impervious surfaces being misclassed as soil is the second most common misclassification of soil and one of the major sources of low soil user accuracy (Table 5). Light (high albedo) impervious surfaces and soil are spectrally and texturally similar and thus easily confused in classification. Light impervious surfaces can include parking lots, paved roads, compacted dirt roads and light-colored roofs. LiDAR intensity may be useful for differentiating soil and impervious surfaces which may be otherwise inseparable in four band optical imagery.

### 4.5. Grass-Tree Confusion

Grass being misclassed as trees is the second most common misclassification of the grass class and one of the major sources of low producer accuracy for grass (Table 5). Grass and tree classes overlap spectrally in imagery, though trees are usually darker and more textured. One common error is the speckling of apparent grass pixels amidst an otherwise continuous tree canopy. Such pixels may be true grass pixels visible through canopy gaps, but commonly they are produced by bright, well-illuminated sun-facing facets of the tree canopy. This effect may be amplified with low sun angle and mixed tree heights. LiDAR height and intensity layers usually help clarify tree-grass confusion but may also overcorrect speckling errors and precipitate additional corrections. If LiDAR is unavailable or inadequate, tree canopy speckle can be reduced using smoothing and majority convolution filters.

### 4.6. Shadows

Shadows are a common source of error in high resolution imagery. A pixel in shadow receives only non-direct sunlight which affects its spectral signature and lowers the signal to noise ratio; with fewer incident photons, fewer reflected photons reach the sensor. Shadows are common at the edges of buildings, shrubs, trees, and forest patches, and in steep topography. Explicably, shadowed features tend to be misclassified among the darker classes: tree, impervious, and water. Shadows on water are commonly misclassified as impervious. Such water errors are sometimes correctable using NHDPlus or NWI layers and by thresholding on low reflectance values in the NIR band.

## 5. Conclusions

We define a classification system for US EPA EnviroAtlas Meter-scale Urban Land Cover (MULC). At 1 × 1 m pixel size, MULC data supports community mapping, planning, modeling and decision making at high spatial resolution as fine as individual trees, buildings and roads. MULC data and more than one hundred sustainability, health, and ecosystem goods and services metrics have been developed for 30 US communities. MULC and other EnviroAtlas data are free and accessible via web browser in EnviroAtlas, as web services, and by download through EnviroAtlas and the EPA Environmental Data Gateway. MULC data are suitable for many applications including tree planting, green infrastructure siting, watershed protection and modeling, urban heat island and stormwater runoff mitigation and mosquito habitat risk mapping. Data and information updates are available at EPA EnviroAtlas. We hope that the guidelines presented here help MULC users and support similar high spatial resolution mapping efforts.

**Author Contributions:** Conceptualization, A.P., K.E., D.R., and G.G.; Formal analysis, A.P., K.E., D.R., and G.G.; Methodology, A.P., K.E., D.R., G.G.; Project administration, A.P., K.E.; Supervision, A.P., K.E.; Validation, A.P.; Visualization, D.R. and G.G.; Writing—original draft, A.P., K.E., D.R., and G.G.; Writing—review & editing, A.P., K.E., D.R., and G.G. All authors have read and agreed to the published version of the manuscript.

**Funding:** This research received no external funding.

**Acknowledgments:** EnviroAtlas is a collaborative effort led by the U.S. Environmental Protection Agency (EPA) Office of Research and Development (ORD) in partnership with the U.S. Geological Survey (USGS), the U.S. Department of Agriculture (USDA), and other federal and non-profit organizations, universities, and communities, including state, county, and city-level stakeholders. EnviroAtlas MULC community data are developed by scientists from the U.S. EPA and associated post-graduate student contractors. MULC team members have included: Jeremy Baynes, Megan Culler, Matthew Dannenberg, Keith Endres, Chelsea Fizer, Gillian Gundersen, Laura Jackson, Akhilesh Khopkar, John Lovette, Megan Mehaffey, Anne Neale, Gwen Oster, Samuel Pardo, Andrew Pilant, Joseph Riegel, Daniel Rosenbaum, Charles Rudder, Alexandra Sears, Shanti Shrestha, Megan Van Fossen, and other EnviroAtlas personnel. The authors thank external collaborators for sharing land cover data: Jarlath O'Neil-Dunne (UVM Spatial Analysis Laboratory), David Diamond (Missouri Resource Assessment Partnership - MoRAP), the Sonoma VegMap Project, Xiaoxiao Li (Central Arizona-Phoenix Long-Term Ecological Research), and the Chesapeake Conservancy. We thank three anonymous reviewers for helpful suggestions.

**Conflicts of Interest:** The authors declare no conflict of interest.

**Disclaimer:** The views expressed in this paper are those of the authors and do not necessarily reflect the views or policies of the U.S. Environmental Protection Agency. Any mention of trade names, products, or services does not imply an endorsement by the U.S. Government or the U.S. Environmental Protection Agency (EPA). The EPA does not endorse any commercial products, services, or enterprises.

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
