# Peer review of "US EPA EnviroAtlas Meter-Scale Urban Land Cover (MULC): 1-m Pixel Land Cover Class Definitions and Guidance"

_remotesensing, doi:10.3390/rs12121909_

Round 1

Reviewer 1 Report

The authors,

Your article on MULC covers the subject of publishing the atlas using NAIP 1m x 1m dataset. The atlas can be used within various disciplines for planning and decision making. The manuscript presents high quality product. I have only two minor observations:

  1. There is inconsistency in use of tense in some parts of the manuscript, which needs to be reviewed, and sentences rephrased. e.g. lines 136, 142, 157-158, 166.
  2. The other is on class definitions, particularly two class assignments. 1. man-made water bodies and natural water bodies could be separated. Also, the assignment of swimming pools and small water surfaces to impervious surface could be re-examined. The other is the agricultural fallow fields, which are assigned to soil class. From my understanding, such fallow fields are covered with wild plants and can be spectral identified with its rough texture than grasses, agriculture or small shrubs. This could be re-considered as part of agriculture, provided there is vegetation cover.

Sincerley,

The reviewer

Author Response

Please see attached PDF. 

Thank you for your time and efforts.

Drew Pilant

Reviewer 2 Report

This paper introduces the definition of MULC product category and its classification by using multi-source data. The content of the article is detailed and logical. There are some questions in this paper as follows:

  1. In 2 Image classification,Combining three classification methods (pixel-based supervised, object-based supervised, and object rule-based) for image classification. Please introduce what specific image segmentation algorithm is used.
  2. In 4 Definitions of MULC Classes of Table2, For the level 2 codes definition of Water, 11 Fresh Water and 12 Salt Water are both water in terms of spectrum. Does the author distinguish only by spectral characteristics or by other auxiliary information?
  3. Table 3 evaluates the accuracy of MULC fuzzy classification in different EnviroAtlas communities. In the classification training process, does each study area select samples individually, or collect all samples for unified training?

Author Response

Please see attached PDF.

Thank you for your time and efforts.

Sincerely,

Drew Pilant

Reviewer 3 Report

In general, this is a well-written manuscript and an entirely competent execution of high-resolution land cover data generation. The authors have done a great job of sort of "following the playbook" of the previous efforts (whose data they use) and addressing the potential pitfalls in generating this type of data. I don't think this paper needs to come back for another round of review, but I do have some minor comments for clarity that I would encourage the authors to address:

1. I don't understand quite well enough how results from three classification methods are combined at the end. I take from the text that all three were run each time, but I assume that there were areas of disagreement between them. How were these differences resolved?
2. Figure 4 + text: I'm confused about the step called "Algorithm Edits". Does this imply that the algorithm was tweaked and then re-run? Because it would seem to me that Figure 4 should include an arrow from "Algorithm Edits" back to "Apply classifier" or similar to illustrate an iterative process of improvement.
3. I felt like I had to really wade into the paper to see which cities were mapped, it would be nice to have a list in the introduction section even if it's a little long.

Author Response

(The authors gave the same response as above.)
